# Investigation on Metabolites in Structural Diversity from the Deep-Sea Sediment-Derived Bacterium *Agrococcus* sp. SCSIO 52902 and Their Biosynthesis

**DOI:** 10.3390/md20070431

**Published:** 2022-06-29

**Authors:** Wenping Ding, Yanqun Li, Xinpeng Tian, Min Chen, Zhihui Xiao, Rouwen Chen, Hao Yin, Si Zhang

**Affiliations:** 1CAS Key Laboratory of Tropical Marine Bio-Resources and Ecology, South China Sea Institute of Oceanology, Chinese Academy of Sciences, Guangzhou 510301, China; dingwenping19@mails.ucas.ac.cn (W.D.); liyanqun20@mails.ucas.ac.cn (Y.L.); xinpengtian@scsio.ac.cn (X.T.); chenmin19@mails.ucas.ac.cn (M.C.); xzh@scsio.ac.cn (Z.X.); gulocrwoo@vip.qq.com (R.C.); 2Southern Marine Science and Engineering Guangdong Laboratory (Guangzhou), Guangzhou 511458, China; 3University of Chinese Academy of Sciences, Beijing 100049, China

**Keywords:** *Agrococcus* sp. SCSIO 52902, metabolites, structural elucidation, biosynthetic pathway

## Abstract

Deep-sea sediment-derived bacterium may make full use of self-genes to produce more bioactive metabolites to adapt to extreme environment, resulting in the discovery of novel metabolites with unique structures and metabolic mechanisms. In the paper, we systematically investigated the metabolites in structurally diversity and their biosynthesis from the deep-sea sediment-derived bacterium *Agrococcus* sp. SCSIO 52902 based on OSMAC strategy, Molecular Networking tool, in combination with bioinformatic analysis. As a result, three new compounds and one new natural product, including 3*R*-OH-1,6-diene-cyclohexylacetic acid (**1**), linear tetradepsipeptide (**2**), *N*^1^,*N*^5^-di-*p*-(*EE*)-coumaroyl-*N*^10^-acetylspermidine (**3**) and furan fatty acid (**4**), together with nineteen known compounds (**5**–**23**) were isolated from the ethyl acetate extract of SCSIO 52902. Their structures were elucidated by comprehensive spectroscopic analysis, single-crystal X-ray diffraction, Marfey’s method and chiral-phase HPLC analysis. Bioinformatic analysis revealed that compounds **1**, **3**, **9** and **13**–**22** were closely related to the shikimate pathway, and compound **5** was putatively produced by the OSB pathway instead of the PKS pathway. In addition, the result of cytotoxicity assay showed that compound **5** exhibited weak cytotoxic activity against the HL-60 cell line.

## 1. Introduction

Deep-sea sediments are considered an important source of structurally diverse secondary metabolites with wide biological activities [1]. To adapt to extreme environments of higher pressure, darkness, lower temperature and lack of oxygen, microorganisms have gradually evolved unique metabolic mechanisms to maintain their living [2]. This means that these microorganisms possess great potential to produce unique and bioactive compounds. In previous studies, some compounds isolated from deep-sea sediments exhibited significant biological properties, such as antimicrobial [3,4], anti-inflammatory [5,6], anti-allergic [7], anti-angiogenesis [8], cytotoxic [9,10,11], and antiviral activities [12]. The genus *Agrococcus* that is a rare actinomycete has been isolated from a wide range of environments [13], and few metabolites from the genus were reported. Only an Antarctic *Agrococcus* strain KRD 186, isolated from sediment core BC043 that was collected at 4060 m below sea level, was found to produce the known glycoglycerolipid GGL3 by using Molecular Networking [14].

As a continuation of our studies on metabolites with structural diversity and biological potential from bacterium isolated from the deep-sea sediment, we started investigating the chemical constitutions and gene information of SCSIO 52902 collected from the western South China Sea at a depth of 2061 meters. The OSMAC (One Strain Many Compounds) strategy [15] was carried out for selecting a better fermentation condition featuring more types of metabolic products. The OSMAC strategy has been shown as a simple and powerful tool that can activate many silent biogenetic gene clusters in microorganisms to make more natural products [16]. In our previous paper, we used OSMAC strategy to find a fermentation condition to specifically produce surfactin compounds [17]. In combination with the Global Natural Products Social (GNPS) Molecular Networking tool [18], we discovered mM20 medium was the better selection for fermentation in large scale, because the ethyl acetate (EtOAc) extract of the fermentation broth not only exhibited more HPLC profiles (Appendix A) but also presented more nodes with red color that wide distributed in different clusters (Figure 1a). Accordingly, a 38 L scale fermentation was carried out in a 65 L fermenter for obtaining enough EtOAc extract for further isolation and purification. This led to the isolation of three new compounds and one new natural product, agrocusin A (**1**), agrotetratide A (**2**), *N*^1^,*N*^5^-di-*p*-(*EE*)-coumaroyl-*N*^10^-acetylspermidine (**3**) and 2-(5-hexylfuran-2-yl)acetic acid (**4**), together with nineteen known ones (**5**–**23**) (Figure 2). The planar structures of compounds **1**–**4** were elucidated by detailed 1D/2D NMR spectroscopy, HRESIMS data and secondary ion mass spectrometry (MS/MS) analysis. The absolute configuration of **1** was confirmed via single-crystal X-ray diffraction analysis (Cu K*α*) and that of **2** was determined by Marfey’s method and chiral-phase HPLC analysis.

Analyzing the structural similarity of the twenty-three compounds, we discovered that some compounds (**1**, **5**, **9** and **13**–**22**) were putatively derived from the shikimate and OSB (*o*-succinylbenzoate) pathway, which was supported by detailed bioinformatic analysis of the genome sequence of SCSIO 52902. In addition, bioactivity screening showed that compound **5** displayed weak antitumor activity against the HL-60 cell line with IC_50_ value of 21.48 μM. This is first research study to systematically examine the metabolites from the genus *Agrococcus* to our knowledge, which provided a chance for obtaining deep insights into the chemical constituents of the genus.

## 2. Results and Discussion

### 2.1. Analysis of the Molecular Networking

To investigate the metabolites in the structural diversity of SCSIO 52902 isolated from deep-sea sediments, eleven different media (Appendix A) were evaluated for metabolites based on OSMAC strategy. The result showed that the four EtOAc extracts from mJNP1A, mMCQ1, mISP4 and mM20 media possessed the more abundant profiles in HPLC-UV chromatogram (Appendix A), and the LC-MS/MS spectra of the four samples were further measured. Molecular Networking (MN) was generated to interconnect the mass spectrometric data from the four samples, and a visual graph (Figure 1a, Appendix A) was drawn using Cytoscape software [19]. MN contained ion peaks from four EtOAc extracts of mJNP1A, mMCQ1, mISP4 and mM20 media, corresponding to green, cyan, indigo and red nodes, respectively. The mM20 medium corresponding to red nodes was selected to ferment in large scale, because the EtOAc extract not only exhibited more abundant HPLC profiles (Appendix A) but also presented wide distributing nodes in different clusters in the MN, meaning that the more diversiform metabolites may be produced. Compared to the shaking flask, the fermenter with sufficient sterile air exhibited more efficient and exhaustive fermentation based on the result of HPLC profiles (Appendix A). In addition, a node of parent ion at *m*/*z* 480.251 in the MN (Appendix A) was identified as compound 3 after further isolation and purification (Figure 1b).

### 2.2. Structural Elucidation

Compound **1** was isolated as a colorless crystal (MeOH-CH_3_COCH_3_). It was assigned the molecular formula C_8_H_10_O_3_ by analysis of its HRESIMS (*m*/*z* 153.0558 [M − H]^−^, calcd for C_8_H_9_O_3_153.0557), suggesting four degrees of unsaturation. The ^1^H NMR spectrum (Table 1) of **1** showed characteristic signals for three olefinic protons at *δ*_H_ 6.16 (H-1, dd, *J* = 9.9, 1.5 Hz), *δ*_H_ 6.13 (H-2, dd, *J* = 10.0, 2.5 Hz), *δ*_H_ 5.66 (H-7, br s), and one oxygenated methine at *δ*_H_ 4.31 (H-3, ddd, *J* = 6.9, 4.9, 2.5 Hz). The ^13^C and DEPT spectra of compound **1** suggested the presence of eight carbon resonances including two methylenes, four methines (one oxygenated and two olefinic carbons), and two quaternary carbons (one olefinic carbon and one carboxyl). The observation of the ^1^H-^1^H COSY correlations of H-1/H-2/H-3/H_2_-4 (*δ*_H_ 2.03 and *δ*_H_ 1.55)/H_2_-5 (*δ*_H_ 3.41 and *δ*_H_ 2.54) disclosed a fragment of -CH=CH-CH(OH)-CH_2_-CH_2_-. The presences of two structural fragments, one six-membered ring and one *α*, *β*, *γ* and *δ*-unsaturated acid, were deduced from HMBC correlations (Figure 3) from H-1, H-2, H_2_-4, H_2_-5 to C-6 (*δ*_C_ 153.6), from H-1, H_2_-5 to C-7 (*δ*_C_ 118.0), and from H-7 to C-8 (*δ*_C_ 170.5). The *E*-geometry of the double bond between C-6 and C-7 was determined by the NOESY correlation of H-1 with H-7 (Figure 4). These above results were unambiguously confirmed by a single-crystal X-ray diffraction analysis using Cu K*α* radiation with a sufficient Flack parameter of −0.04 (11) (CCDC 2164734) (Figure 4), resulting in the elucidation of the configuration of C-3 as *R*. Moreover, compound **1** was named agrocusin A.

Compound **2** was obtained as a white powder. Its molecular formula was determined to be C_22_H_40_N_2_O_7_ with four degrees of unsaturation on the basis of HRESIMS data (*m*/*z* 445.2902 [M + H]^+^, calcd for C_22_H_41_N_2_O_7_ 445.2908). The ^13^C and DEPT spectra (Table 2) displayed four carboxyl carbons at *δ*_C_ 171.4~177.9, four *α*-carbons at *δ*_C_ 52.5~80.1, two methylene carbons at *δ*_C_ 41.9 and *δ*_C_ 45.0, four methine carbons at *δ*_C_ 25.6~32.0, and eight methyl carbons at *δ*_C_ 17.8~24.0. The above-mentioned information suggested that compound **2** was analogous to linear tetradepsipeptide [20]. Four structural fragments of Hmv (2-hydroxy-4-methylvaleric acid), Val (valine), Hiv (2-hydroxyisovaleric acid) and Leu (leucine) were deduced by detailed analyses of ^1^H-^1^H COSY and HMBC spectra (Figure 3). The linear structure of Hmv-Val-Hiv-Leu was constructed by HMBC correlations from H*_α_*-Leu (*δ*_H_ 4.43, dd, *J* = 10.3, 4.4 Hz) to CO-Hiv (*δ*_C_ 171.4), from H*_α_*-Hiv (*δ*_H_ 4.86, overlapped) to CO-Val (*δ*_C_ 172.3), and from H*_α_*-Val (*δ*_H_ 4.46, d, *J* = 6.1 Hz) to CO-Hmv (*δ*_C_ 177.9). The planar structure was further confirmed by tandem ESI-MS/MS fragment peaks at *m*/*z* 331.22, 314.20, 232.16, 214.14 and 186.15 (*m*/*z* 445.2818 as precursor) (Appendix A). To determine the absolute configuration of compound **2**, Marfey’s method and chiral HPLC analysis were implemented. In brief, compound **2** (0.25 mg) was hydrolyzed with 1 mL 6 M HCl at 110 °C for 18 h, and then the hydrolysate dried was derivatized with N*α*-(2,4-Dinitro-5-fluorophenyl)-l-alaninamide (l-FDAA) and it was subsequently analyzed by HPLC using a YMC-Pack Ph column [17] (Appendix A). The absolute configurations of Val and Leu were unambiguously determined as D and L, respectively, by comparison with the retention times of the standard amino acid derivatives (Table 3). Furthermore, the absolute configurations of Hmv and Hiv residues were determined by analyzing the hydrolysate of **2** on a ligand-exchange type chiral-phase HPLC column (Sumichiral OA-5000L) by comparison with commercially available standard *α*-hydroxy acids. The Hmv and Hiv structures from the hydrolysate were eluted at 31.81 and 17.65 min, respectively, which were consistent with the standards D(*R*)-Hmv (*t*_R_ = 32.57 min) and L(*S*)-Hiv (*t*_R_ = 18.12 min), respectively (Table 3, Appendix A). Therefore, the structure of compound **2** was confirmed and named agrotetratide A.

Compound **3** was obtained as a colorless oil and possessed the molecular formula C_27_H_33_N_3_O_5_, as deduced by the molecular ion peak at *m*/*z* 480.2492 [M + H]^+^ (calcd for C_27_H_34_N_3_O_5_ 480.2493), indicating thirteen degrees of unsaturation. Analyzing the ^13^C spectrum (Appendix A) revealed that most carbon signals appeared in pairs with an approximate height ratio of 1:1. Two HPLC profiles were apparently observed by further HPLC analysis (Appendix A). After purifying the large HPLC profile, the NMR spectra (Appendix A) were measured again. The carbon signals also appeared in pairs consistent with the above result, suggesting that the molecule was unstable and easily changeable. The speculation was confirmed by observing two profiles in HPLC analysis again (Appendix A). NMR data (Table 2) suggested that compound 3 was analogous to spermidine [21,22]. Together with molecular formula C_27_H_33_N_3_O_5_, detailed analyses of ^13^C and DEPT spectra suggested that compound **3** included one methyl, seven methylenes, two pairs of double bonds, two benzenes and three amides. The para-position of benzene was replaced by a hydroxyl group, resulting in the chemical shift value of approximately 160 ppm. These structural fragments of two *p*-coumaroyl, one acetyl, one 1,3-propanediamine and one 1,4-butanediamine were disclosed by detailed analyses of HMBC and ^1^H-^1^H COSY spectra (Figure 3). The planar structure was established by HMBC correlations from H_2_-2 to C-1’, from H_2_-4 to C-1”, C-6, from H_2_-6 to C-1”, C-4, and from H_2_-9 to C-1′′′. Corresponding to the above results, ESI-MS/MS showed characteristic fragment peaks at *m*/*z* 334.21, 316.20, 275.18, 204.10, 188.18 and 147.04 (*m*/*z* 480.2358 as precursor) (Appendix A). The type of spermidine was thought to be photoisomerization because the configuration of double bond was easily changed after being lighted [22]. Compound **3** can theoretically produce four optic isomers due to two pairs of double bonds. After being placed two months at room temperature in a light context, compound **3** was isomerized and further resulted in four monomers after conducting HPLC analysis (Appendix A). In the HPLC chromatogram, the HPLC profile of the *EE*-configuration was the highest and that of the *ZZ*-configuration was lowest.

Compound **4** was obtained as a white powder and assigned the molecular formula C_12_H_18_O_3_ with four degrees of unsaturation, according to the negative HRESIMS ion at *m*/*z* 209.1188 [M − H]^−^ (calcd for C_12_H_17_O_3_ 209.1183). The ^1^H NMR spectrum (Table 1) showed two olefinic protons at *δ*_H_ 6.12 (H-4, d, *J* = 3.0 Hz) and *δ*_H_ 5.91 (H-5, d, *J* = 3.1 Hz). The ^13^C and DEPT spectra (Table 1) of compound **4** indicated the presences of twelve carbon resonances, including one methyl, six methylenes, two methines (two olefinic carbons), and three quaternary carbons (two olefinic carbons and one carboxyl). The fragment of *α*-disubstituted furan was revealed by analyzing the NMR data of C-6 (*δ*_C_ 156.5), C-3 (*δ*_C_ 145.0), C-4 (*δ*_C_ 108.8), and C-5 (*δ*_C_ 105.5) and corresponding spin–spin coupling protons at *δ*_H_ 6.12 and *δ*_H_ 5.91. The speculation was confirmed by 2D NMR spectra (Appendix A). In addition, two substitutions were assigned as acetic acid and hexyl according to HMBC correlations from H-2 to C-1, C-3 and C-4 and from H-7 to C-5 and C-6, together with ^1^H-^1^H COSY correlations of H-7/H-8/H-9/H-10/H-11/H-12 (Figure 3). Compound **4** was a new natural product and was firstly synthesized by the Whitehead R. C. group in 2001 [23], namely 2-(5-hexylfuran-2-yl)acetic acid.

Compound **5** was elucidated as a naphthoquinone linked with homocysteine by comparing its NMR data with those values reported in the literature [24], namely agrocuquinone A. It was firstly discovered and patented as a purple dye combined with its analogs by Japanese researchers in 1977 because of its strong purple color. It was the second time that the molecular structure of **5** was reported, to the best of our knowledge. Compounds **6**–**13** were identified as cyclodipeptides, in which compounds **6**–**9** were elucidated by NMR data [25,26] and single-crystal X-ray diffraction analysis using Cu K*α* radiation (Figure 5). Compounds **10**–**13** were determined by comparing their NMR data and specific rotations with those values reported in the literature [27,28,29,30], and verified by Marfey’s analysis (Appendix A). Compounds **14**–**17** were elucidated as phenylbutane derivatives, in which configurations of the hydroxyl groups were confirmed by comparing their NMR data and specific rotations with those values reported in the literature [31,32,33]. Compounds **18**–**20** were elucidated as simple phenylethane derivatives, and their structures were identified by analyzing their NMR data. Compounds **21**–**23** were identified as anthranilic acid, indole-3-acetic acid and *N*-isobutylacetamide by analyzing their NMR data, respectively.

In addition, to discuss the putative biosynthetic pathway of isolated compounds, we initially classified them into five categories based on their structural similarity. The first group was related to the shikimate pathway, including compounds **1**, **3** and **14**–**22**; the second group was related to cyclodipeptide synthesis, including compounds **6**–**13**; the third group was related to the acetate-malonate pathway, including compounds **4** and **5**; the fourth group was related to acetylation, including compounds **3** and **23**; the last group was related to a depsipeptide assembly, which only contained compound **2**.

In order to gain insight into the relationship between metabolic products and biosynthetic gene clusters (BGCs), the genomic DNA of the strain SCSIO 52902 was sequenced by using Oxford Nanopore Technologies (ONT) technology [34], and the final genome assembly (GenBank CP095269) provided one circular contig containing 2,693,362 bp with a GC content of 73.17% (Appendix A).

### 2.3. Putative Biosynthetic Pathway

To identify the BGCs related to all isolated metabolites in the genomic sequence of the strain, we uploaded the full sequences of genome to antiSMASH bacterial version 6.0.1 [35] with the default “relaxed strictness” settings. The software predicted five BGCs (regions 1.1–1.5) located on the circular chromosome (Appendix A), corresponding to terpene, T3PKS, oligosaccharide, betalactone and redox-cofactor types. Given that some compounds were considered as primary metabolites by antiSMASH, such as amino acid and fatty acid, we once again retrieved BGCs from the software by changing the settings to “loose strictness”. The output displayed extra fourteen biosynthetic regions (Appendix A) that contained two related fatty acid synthesis BGCs (regions 2.2 and 2.17, Figure 6). Moreover, one BGC (region 2.1, Figure 6) that contained the core genes of glyoxylate cycle [36] and a 5-enol pyruvylshikimate-3-phosphate biosynthesis gene related to shikimate pathway [37] was observed. The biosynthesis of compound **4** was putatively related to fatty acid BGCs, but we failed to discover an enzyme similar to RSP_1091 (UfaO) [38], a reported enzyme that can catalyze the conversion of 19M-UFA to 19Fu-FA resulting in a furan ring. Interestingly, a P450 gene and some oxidoreductase genes were discovered in region 2.2 (Figure 6). In combination with the literature reporting that furan ring could be produced by cytochrome P450 enzyme [39,40], we proposed that the biosynthesis of compound **4** was related to region 2.2.

In addition, the formation of compound **5** was putatively associated with two gene clusters, region 1.2 (T3PKS type, Figure 6) and region 1.4 (betalactone type, Figure 6). However, detailed bioinformatic analysis showed that the core gene in region 1.2 shared less than 30% identity with RppA [41], a type III polyketide synthase that can catalyze the formation of 1,3,6,8-tetrahydroxynaphthalene (THN). In addition, we did not find the reductase gene that can reduce phenolic hydroxy from region 1.2. Meanwhile, 1,4-naphthoquinones (1,4-NQs) were also derived from the shikimate pathway [42], such as vitamin K_2_. Coincidentally, some compounds were also related to the shikimate pathway based on structural analysis, including compounds **1**, **3** and **14**–**22.** In region 1.4 (Figure 6), we discovered a 3-deoxy-d-arabinoheptulosonate 7-phosphate (DAHP) synthase [43] that was the first enzyme in the shikimate pathway. Moreover, 3-phosphoshikimate 1-carboxyvinyltransferase [44] (EPSP synthase) that can catalyze the transfer of the enolpyruvyl moiety from phosphoenolpyruvate (PEP) to the 5-hydroxyl of shikimate-3-phosphate (S3P) to form 5-enol pyruvylshikimate-3-phosphate (EPSP) and then further yields chorismate was also discovered from region 2.1 (Figure 6). In addition, we investigated the annotation information of the genome sequence of the strain using general database annotation, including GO, KEGG, Pfam, SwissProt and TrEMBL. We searched almost all core enzymes (Appendix A) related to the shikimate and OSB pathway [45] against the resulting annotation information, finding PchA [46], which was functionally similar to MenF [42], MenA-E [42] and MenG [42] (Appendix A). These results strongly supported the putative biosynthetic pathway as depicted in Figure 1. Putatively, compound **5** was further synthesized by two substrates of 1,4-dihydroxy-2-naphtoate and l-homocysteine, the latter from *S*-adenosyl-l-homocysteine that was produced by a class I SAM-dependent methyltransferase discovered from region 1.4 (Figure 6) catalyzing *S*-adenosyl-l-methionine.

Compounds **14**–**17** were putatively derived from *O*-succinylbenzoate by decarboxylation and oxidoreduction. In the tryptophan (Trp) biosynthetic pathway, compounds **21** and **22** were produced by anthranilate synthase component 2 (TrpG) [47] and subsequent decarboxylase, respectively. Meanwhile, compound **1** structurally possessed the 3*R*-hydroxy configuration, which was consistent with prephenate, suggesting that compound **1** may well be derived from prephenate through reduction and decarboxylation. Compound **20** was isolated with more than 200 mg, and it was putatively produced by the decarboxylation of phenylpyruvic acid. Compounds **18** and **19** were putatively derived from either 4-hydroxyphenylpyruvic acid or compound **20**. Compounds **9** and **13** were putatively produced via reaction phenylalanine (Phe) derived from shikimate pathway with proline (Pro) or valine (Val). Similarly, six other cyclodipeptides were putatively synthesized by cyclodipeptide synthase [48]. Excitingly, in region 1.5, we found a gene coding tRNA synthetases class I (I, L, M and V) that can synthesize aminoacyl-tRNA, and aminoacyl-tRNA can be used as substrate to synthesize the two peptide bonds of various cyclodipeptides by cyclodipeptide synthases [49] (Figure 6). Meanwhile, the tRNA synthetases class I was thought to be related to the biosynthesis of compound **2** that was putatively assembled by non-ribosomal peptide synthase. Compounds **3** and **23** were considered as acetylation products, and compound **3** was putatively produced by reaction one spermidine with two *p*-coumaroyl derived from Phe or Tyr from the shikimate pathway and one acetyl.

### 2.4. Biological Activities

Compounds **1**–**17** were tested for cytotoxicity against A-549, HL-60 and HCT-116 human tumor cell lines at 30 μΜ concentration by CCK-8 assay [50]. The result showed that only compound **5** exhibited weak antitumor activity against HL-60 with 77.12% inhibition rate (Appendix A). Subsequently, the IC_50_ value of compound **5** against HL-60 was further measured by CCK-8 assay in a gradient descent manner, resulting in 21.48 μM (Staurosporine as positive control with IC_50_ value of 0.03782 μM). In addition, all isolated compounds were tested for antibacterial activity, resulting in the fact that none of all compounds displayed obvious antibacterial activity.

## 3. Materials and Methods

### 3.1. General

The general experimental procedures were described in our previous paper [17].

### 3.2. Microorganism and Growth Conditions

SCSIO 52902 was isolated and purified from sediment collected from the western South China Sea at a depth of 2061 meters. Analysis of the 16S rRNA sequence of SCSIO 52902 indicated that the bacterium was a member of the *Agrococcus* sp. and the bacterium shared 99.05% identity with *Agrococcus lahaulensis* DSM 17612(T) (GenBank accession no. AULD01000008). Initially, the strain was cultivated in eleven different liquid media (Appendix A). mM20 liquid medium was selected for fermentation on a 38 L scale based on the results of HPLC profiles and Molecular Networking. The fermentation experiment was carried out in a 65 L fermenter (Bioflo 610, Eppendorf, Germany) at 28 °C with the stirring rate at 90-135 rpm and 20% dissolved oxygen (DO). Samples were taken to analyze HPLC profiles at 23 h, 48 h, 72 h, 96 h and 115 h (Appendix A). Fermentation was stopped after 119 h of cultivation according to HPLC analysis results, and variations in both pH and DO of the medium were observed (Appendix A).

### 3.3. Complete Genome Sequence and Bioinformatic Analysis

High molecular weight genomic DNA (gDNA) of SCSIO 52902 was extracted according to Oxford Nanopore Technologies (ONT) protocol [34], and size selection was carried out by automatic BluePippin system. Then, library preparation was performed with the SQK-LSK109 kit. The final genome assembly provided one circular contig with 2,693,362 bp with a GC content of 73.17% after using Canu v1.5 for assembly, Racon v3.4.3 for rectification, Circlator v1.5.5 for cyclization and Pilon v1.22 for correction. All above software used with default parameters. Genome sequence has been deposited in GenBank under accession number CP095269. Genome sequence annotation was performed using general database, including GO, KEGG, Pfam, SwissProt and TrEMBL, and the potential secondary metabolite BGCs were predicted by antiSMASH 6.0.1 [35].

### 3.4. Extraction and Isolation

The entire fermentation broth (38 L) was extracted with an equivalent volume of EtOAc three times at room temperature. The EtOAc layer was separated from the aqueous phase and concentrated using a rotary evaporator in vacuo to obtain the dry extract (22 g). Subsequently, the EtOAc extract was separated on MPLC-C-18 with MeOH/H_2_O (10:90, 20:80, 30:70, 40:60, 50:50, 60:40, 70:30, 80:20, 90:10 and 100:0) to obtain Fr.1-Fr.10. Fr.3 (1 g) was chromatographed on Sephadex LH-20 column eluted with MeOH to afford nine subfractions Fr.3.1-Fr.3.9. Fr.3.3 (120 mg) was further purified by semi-preparative HPLC to afford **1** (1.4 mg), **18** (14.8 mg) and **19** (5.0 mg). Fr.3.1 (200 mg) was repeatedly purified by semi-preparative HPLC to afford **6** (6.8 mg), **7** (20.4 mg), **8** (11.6 mg), **10** (1.0 mg), **11** (1.7 mg), **12** (1.6 mg) and **23** (8.4 mg). Fr.3.5 (17 mg) was purified by semi-preparative HPLC to afford **21** (5.9 mg). Fr.4 (1.3 g) was also chromatographed on Sephadex LH-20 column eluted with MeOH to afford nine subfractions Fr.4.1-Fr.4.9. Fr.4.5 (800 mg) was repeatedly purified by semi-preparative HPLC to afford **9** (12.4 mg), **13** (2.4 mg), **14** (2.3 mg), **15** (2.3 mg), **16** (11.7 mg) and **20** (200 mg). Fr.5 (1.6 g) was also chromatographed on Sephadex LH-20 column eluted with MeOH to afford nine subfractions Fr.5.1-Fr.5.9. Fr.5.4 (80 mg), Fr.5.5 (17 mg) and Fr.5.9 (102 mg) were further purified by semi-preparative HPLC to afford **3** (3.5 mg), **22** (3.8 mg) and **5** (1.6 mg), respectively. Fr.6 (1.3 g) was chromatographed on Sephadex LH-20 column eluted with MeOH to afford thirteen subfractions Fr.6.1-Fr.6.13. Fr.6.7 (10 mg) was further purified by semi-preparative HPLC to afford **17** (2.5 mg). Fr.7 (1 g) was also chromatographed on Sephadex LH-20 column eluted with MeOH to afford eleven subfractions Fr.7.1-Fr.7.11. Fr.7.2 (41 mg) and Fr.7.5 (43 mg) were further purified by semi-preparative HPLC to afford **2** (1.4 mg) and **4** (1.8 mg), respectively.

Agrocusin A (**1**): colorless crystal (MeOH-CH_3_COCH_3_), m.p. = 130.0~130.7 °C, [*α*]25D +68.5 (c 0.1, MeOH), UV (MeOH) *λ*_max_ (log *ε*) 259 (4.06) nm, 197 (3.55) nm. IR (film) *ν*_max_ 3311, 2945, 2833, 1683, 1647, 1456, 1396, 1111, 1020 and 682 cm^−1^. HRESIMS *m*/*z* 153.0558 [M − H]^−^ (calcd for C_8_H_9_O_3_153.0557). ECD (MeOH): 263 nm (Δ*ε* = 0.29), 252 nm (Δ*ε* = 0.32), 213 nm (Δ*ε* = −0.24). ^1^H NMR (CD_3_OD, 700 MHz) and ^13^C NMR (CD_3_OD, 176 MHz), see Table 1 and Appendix A.

Agrotetratide A (**2**): white powder, [*α*]25D +6.0 (c 0.1, MeOH), UV (MeOH) *λ*_max_ (log *ε*) 203 (3.82) nm. IR (film) *ν*_max_ 3319, 2945, 2833, 1734, 1653, 1541, 1456, 1396, 1114, 1020, and 665 cm^−1^. HRESIMS *m*/*z* 445.2902 [M + H]^+^ (calcd for C_22_H_41_N_2_O_7_ 445.2908). ECD (MeOH): 219 nm (Δ*ε* = 0.16), 202 nm (Δ*ε* = −0.10). ^1^H NMR (CD_3_OD, 700 MHz) and ^13^C NMR (CD_3_OD, 176 MHz), see Table 2 and Appendix A.

*N*^1^,*N*^5^-di-*p*-coumaroyl-*N*^10^-acetylspermidine (**3**): colorless oil, UV (MeOH) *λ*_max_ (log *ε*) 310 (4.37) nm, 298 (4.36) nm, 224 (4.19) nm, 204 (4.22) nm. IR (film) *ν*_max_ 3309, 2945, 2835, 1653, 1558, 1541, 1516, 1456, 1109, 1018, 831, and 675 cm^−1^. HRESIMS *m*/*z* 480.2492 [M + H]^+^ (calcd for C_27_H_34_N_3_O_5_ 480.2493). ^1^H NMR (CD_3_OD, 700 MHz) and ^13^C NMR (CD_3_OD, 176 MHz), see Table 2 and Appendix A.

2-(5-hexylfuran-2-yl)acetic acid (**4**): white powder, UV (MeOH) *λ*_max_ (log *ε*) 204 (3.01) nm. IR (film) *ν*_max_ 3323, 2947, 2835, 1660, 1448, 1417, 1114, 1018, and 667 cm^−1^. HRESIMS *m*/*z* 209.1188 [M − H]^−^ (calcd for C_12_H_17_O_3_ 209.1183). ^1^H NMR (CDCl_3_, 700 MHz) and ^13^C NMR (CDCl_3_, 176 MHz), see Table 1 and Appendix A.

Agrocuquinone A (**5**): purple powder, [*α*]25D +16.4 (c 0.001, MeOH), HRESIMS *m*/*z* 290.0483 ([M + H]^+^, calcd for C_14_H_12_NO_4_S 290.0482). 

Cyclo(l-Pro-l-Val) (**6**): colorless crystal (MeOH), m.p. = 121.6~122.1 °C, [*α*]25D −134.4 (c 0.1, MeOH), Flack parameter = 0.01 (**7**). 

Cyclo(l-Pro-l-Leu) (**7**): colorless crystal (MeOH), m.p. = 91.3~92.2 °C, [*α*]25D −100.7 (c 0.1, MeOH), Flack parameter = 0.02 (**15**). 

Cyclo(l-*trans*-Hyp-l-Leu) (**8**): colorless crystal (MeOH), m.p. = 101.6~102.0 °C, [*α*]25D −104.9 (c 0.1, MeOH), Flack parameter = −0.09 (**9**). 

Cyclo(l-Pro-l-Phe) (**9**): colorless crystal (MeOH), m.p. = 127.4~127.9 °C, [*α*]25D −92.9 (c 0.1, MeOH), Flack parameter = −0.03 (**9**).

Cyclo(d-Pro-l-Ile) (**10**): white powder, [*α*]25D +25.9 (c 0.1, MeOH). Cyclo(l-*trans*-Hyp-l-Ile) (**11**): white powder, [*α*]25D −71.8 (c 0.1, MeOH). Cyclo(l-Ala-l-Ile) (**12**): white powder, [*α*]25D −15.7 (c 0.1, MeOH). Cyclo(l-Val-l-Phe) (**13**): white powder, [*α*]25D −5.7 (c 0.1, MeOH). The retention times of Marfey’s derivatives of **8** and **10**−**13** were listed in Appendix A.

(2*S*,3*S*)-1-phenyl-2,3-butanediol (**14**): white powder, [*α*]25D +7.2 (c 0.1, MeOH). (2*R*,3*S*)-1-phenyl-2,3-butanediol (**15**): white powder, [*α*]25D −11.2 (c 0.1, MeOH). (*S*)-3-hydroxy-4-phenylbutanoic acid (**16**): white powder, [*α*]25D +5.9 (c 0.1, MeOH).

Crystallographic data for the structures of **1** and **6**–**9** have been deposited in the Cambridge Crystallographic Data Centre (deposition number CCDC 2164734, 2164793, 2164752, 2164753 and 2164750). Copies of the data can be obtained free of charge from the CCDC via www.ccdc.cam.ac.uk (accessed on 6 April 2022).

### 3.5. Molecular Networking

The method was described in our previous work [17]. In brief, a 25 μL aliquot (5 mg/mL, dissolved in MeOH) of the samples was analyzed by LC-MS/MS on a YMC-Pack ODS-A (250 mm × 4.6 mm, S-5 μm, 1 mL/min) column eluted with a gradient program of CH_3_CN/H_2_O (0.1% formic acid modifier): 10% to 100% in 30 min, then isocratic elution to 40 min, at a flow rate of 1 mL/min, during which the mass spectrometer was set to detect *m*/*z* 50–1500 in the positive ESI mode and with an automated full dependent MS/MS scan enabled. The chromatogram was converted digitally to a .mzML file using freely available MSConvert software and, subsequently, submitted to the GNPS website. Molecular Networking was generated to interconnect MS/MS data from the four samples.

### 3.6. Marfey’s Analysis

As described in our previous paper [17], 0.25 mg amounts of compounds **2**, **8** and **10**–**13** were hydrolyzed with 6 M HCl (1 mL) at 110 °C for 18 h, and the reaction mixture was cooled to room temperature and evaporated to dryness. The resulting residue was diluted with 100 μL of water and then treated with 80 μL of 1% acetone solution of l-FDAA and 40 μL of 1 M NaHCO_3_ at 40 °C for 1 h. Subsequently, the reaction was quenched by the addition of 2 M HCl (20 μL), followed by vaporization in vacuo and then dilution with MeOH (200 μL). After filtration through a 0.45 μm syringe filter, a 25 μL aliquot was injected into HPLC for measuring retention times under analytical conditions as follows: column: YMC-Pack Ph (250 mm × 4.6 mm, S-5 μm); A phase: ultrapure water; B phase: CH_3_CN; C phase: aqueous solution of 0.1% formic acid; gradient program 1: 0 min (42%A–28%B–30%C) to 20 min (42%A–28%B–30%C) to 50 min (30%A–40%B–30%C) to 53 min (30%A–40%B–30%C); gradient program 2: 0 min (60%A–20%B–20%C) to 20 min (52%A–28%B–20%C) to 50 min (37%A–43%B–20%C) to 53 min (37%A–43%B–20%C); flow rate: 1 mL/min; detection: UV 340 nm. The chiral configurations of hydrolysates were determined by comparison with retention times of authentic amino acid standards that were previously l-FDAA derivatized.

### 3.7. Chiral HPLC Analysis

Compound **2** (0.25 mg) was hydrolyzed with 6 M HCl (400 μL) at 110 °C for 7 h, and the reaction mixture was dried by evaporation. The residue was dissolved in 80 μL water, and then a 20 μL aliquot was subjected to HPLC analysis (Sumichiral OA-5000L, 4.6 × 150 mm; 15% 2-propanol in 2 mM CuSO_4_ aqueous solution; 1 mL/min; UV 254 nm). The retention times for the standards were 11.144 min for (*R*)-Hiv, 18.122 min for (*S*)-Hiv, 32.568 min for (*R*)-Hmv, and 41.889 min for (*S*)-Hmv, while the hydrolysate of **2** displayed HPLC profiles at 17.651 min and 31.811 min (Appendix A).

### 3.8. Cytotoxicity Assay

Three human tumor cell lines A-549, HL-60 and HCT-116 were used in the cytotoxicity assay, in which cytotoxic activity was evaluated by WST-8 reagent [50]. The experimental method was referenced in our previous paper [17]. In brief, the three cells were cultured in 96-well plate in 2000 cells/well with 80 μL overnight in a 5% CO_2_ incubator at 37 °C. The test compounds **1**–**17** (30 μM and 20 μL) or Staurosporine (10 μM and 20 μL) as positive control were added in wells, and the plates were incubated under the dark condition for 72 h. Then, cells were treated with the WST-8 reagent in a 5% CO_2_ incubator at 37 °C for 1 h. Optical density was measured on an EnVision spectrophotometer at 450 nm and the inhibition rate was calculated as the equivalent of the following: inhibition rate % = (OD_S_ − OD_NC_)/(OD_STSP_ − OD_NC_) × 100%, where OD_S_, OD_NC_ and OD_STSP_ were the absorption values of well with additional test compound, DMSO as negative control and Staurosporine, respectively. All measurements were performed in triplicate. Then, the IC_50_ value of compound **5** against HL-60 cell line was further measured by a CCK-8 assay in a gradient descent manner using Staurosporine as the positive control.

### 3.9. Antibacterial Assay

Antibacterial evaluation against four indicator bacteria (*B*. *subtilis*, *B*. *thuringiensis*, *S*. *aureus* and *E*. *coli*) were carried out by the agar well diffusion method [51]. Gentamicin sulfate was used as the positive control. Briefly, seeded agar was made using LB agar medium. Approximately at 40~45 °C, 0.2 mL of seeded inoculum was added in the 100 mL medium, and then it was poured into the sterilized petri dishes under aseptic conditions. After solidification, wells of 6 mm diameter were punched into the agar medium. These agar wells were added with either 2 mg/mL 5 μL test compounds or solvent methanol as the negative control. The plates were incubated at 37 °C for 24 h. Antibacterial activity was evaluated by measuring the zone of inhibition against indicator bacteria.

## 4. Conclusions

In this paper, we initially used OSMAC strategy to activate silent gene clusters, and then we used Molecular Networking tool to make the results visual so as to screen for fermentation condition. Consequently, twenty-three compounds were isolated and identified, including three new compounds (**1**–**3**) and one new natural product (**4**), from the EtOAc extract via extensive spectroscopic analysis, single-crystal X-ray diffraction, Marfey’s method and chiral-phase HPLC analysis. Compound **2** was a rarely natural tetradepsipeptide, possessing linear structure comprising d(*R*)-Hmv, d(*R*)-Val, l(*S*)-Hiv and l(*S*)-Leu and connected via amide, ester and amide linkages, respectively. Compound **3** was identified as photoisomerization, which was detected in Molecular Networking, yet it failed to be annotated by GNPS. The structures possessing two *p*-coumaroyl similar to compound **3** were discovered from bacteria for the first time, which were generally isolated from plant. Compound **4** was a relatively rare 12-carbon length fatty acid featuring a *α*-disubstituted furan moiety without methyl in *β*-position. The furan fatty acids acted as powerful radical scavengers in defense against oxidative stress [52], suggesting that compound **4** possibly was produced under the stress of sufficient oxygen in fermenter. In addition, we discovered that fourteen compounds (**1**, **3**, **5**, **9** and **13**–**22**) were putatively related to the shikimate pathway according to structural analysis in combination with bioinformatic analysis, and we further proposed a plausible biosynthetic pathway. The above results suggested that it was worth to keep looking at deep-sea sediment bacteria strains for the sake of novel structures and uncommon biosynthetic pathways. Meanwhile, the whole-genome sequencing result showed the SCSIO 52902 strain possessed smaller genome sequence about 2.69 Mbp, and five BGCs were predicted by using antiSMASH bacterial version 6.0.1 with standard settings. Moreover, we found that all BGCs possessed a lower similarity to known clusters with a maximum value of 50%. These results suggest that the strain may need make full use of the limited genes to produce more metabolites to maintain its life and to compete or communicate with other species in surroundings. In addition, the cytotoxicity assay showed that only compound **5** displayed weak antitumor activity against the HL-60 cell line with an IC_50_ value of 21.48 μM, together with the result of antibacterial assay, suggesting that biological functions of isolated metabolites could be noncompetitive.

## Data Availability

The data presented in this study are available in this article and Appendix A.

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
