# Peer review of "Investigation on Metabolites in Structural Diversity from the Deep-Sea Sediment-Derived Bacterium Agrococcus sp. SCSIO 52902 and Their Biosynthesis"

_marinedrugs, 2022, doi:10.3390/md20070431_

Round 1

Reviewer 1 Report

This manuscript described the metabolites in structurally diversity and corresponding genes information from the deep-sea sediment-derived bacterium Agrococcus sp. SCSIO 52902 based on OSMAC strategy, Molecular Networking tool, and bioinformatic analysis. Three new compounds were isolated. Bioinformatic analysis revealed compounds 1, 3, 9 and 13-22 were closely related to shikimate pathway, and compound 5 was putatively derived from OSB pathway rather than polyketide synthase pathway. The results and the methods used in this research are interesting and the manuscript is organized well. Few suggestions are listed below.

1.     The title of table 2, change “1 and 3” to “2 and 3”.

2.     Figure 1. Graph (b). please show the whole MS2 spectrum. The MW of compound 3 is 480.25 but what you display is 334.21.

3.     In biological activities part. Has the sample been secondary screened for bioactivity? Moreover, please show the result of control drug.

Reviewer 2 Report

In this manuscript Ding and collaborators present chemical research performed on a deep-sea sediment-derived bacterium Agrococcus sp which was isolated from 2061 m in the South China Sea. The work is interesting as the authors use various strategies to explore all chemical diversity present in the strain. This effort in itself is very interesting, however, the manuscript writing doesn't succeed in showing the importance of this approach, i.e. GNPS, traditional structural elucidation and cultivation, and genomics. English revisions should be made. I have highlighted in yellow all phrases I have found to contain English mistakes which, unfortunately, can make the manuscript loose the enthusiasm that the results should provide to the reader. Some information should be included on why it is important to find these compounds in the strain. Also, information on previous compounds found in this bacterial genus should be added in the introduction to highlight importance of new findings or to expand on previously published results.

The hypothesis the work is based on states that deep-sea sediment bacteria should be making novel compounds due to their exposure to an extreme environment (lines 36-38), however, no further mention of this was found in the manuscript. Did the authors find the novelty they were expecting? are these compounds novel enough? Conclusions look more like a condensed list of the results, it would greatly benefit the manuscript if a discussion on the importance of using different strategies to study all possible methods on one strain is included in the manuscript. This way, the conclusions could be based on accepting or rejecting the initial hypothesis. Is it worth it to keep looking at deep-sea sediment bacteria strains? or is it the same as looking at strains from the shore? Did the authors find anything regarding this in their results?

I will upload the version with my comments and highlights (English revision suggestions) for the authors to be able to easily follow them.

Reviewer 3 Report

The authors of this paper have done a lot of work from the selection of the
medium according to the OSMAC strategy to the isolation,
structure elucidation including absolute configurations of some compounds
and biological activity results. But, the isolated compounds have a low
novelty and only one compound 5 had a weak cytotoxic activity.
So, in my opinion the results suggest that Deep-Sea Sediment-Derived
Bacterium Agrococcus sp. SCSIO 3 52902 are not perspective source of
bioactive compounds.

Line 122. «While the configuration of 3-hydroxy was elucidated as R». It is not correct sentence, the correct “…the configuration of C-3 was elucidated as R” or “ … 3R…”. But it is not clear which method? Only using Flack parameter data without Mosher’s method?

The compounds have not any names, trivial or IUPAC. Compound 1, 2, 3 etc it is not correct, especially for known compounds.

Table 1. for compound 4 - 3.67 (s) should be (2H), and the means proton signals for H2-7 – H2-11 should be (2H) too in the table.

Table 2. In the head the numbers of compounds 2 and 3 should calibrate to center like in the table 1.

Line 291. “Compounds 1-17 were tested for cytotoxicity against A-549, HL-60 and HCT-116 human tumor cell lines at 30 μΜ concentration by CCK-8 assay [47]”. What about cytotoxicity against normal cells? If such data is already given in the literature for compound 5, a reference should be given.

Why you given % inhibition rate and not IC50? The compound 5 was isolated 1.6 mg.

In my opinion, little biological activity has been done. It would be nice to add additional data on biological activity

Conclusions.

“Initially, the fermentation condition was screened by using OSMAC strategy and Molecular Networking tool, and 38 L fermentation broth was cultivated in a 65 L fermenter”. why is this information in the Conclusions?

“Besides, bioactive assay presented that only compound 5 displayed weak antitumor activity against HL-60 with 77.12% inhibition rate, suggesting that biological functions of isolated metabolites need to be further studied” This conclusion is weakly justified, moreover, as I say earlier, why you give the results not IC50?

Round 2

Reviewer 3 Report

Dear authors, you have done a great job of making changes to the manuscript.